# RetPur: Diffusion Purification Model for Defending Hash Retrieval Target Attacks

## Abstract

Deep Neural Networks (DNNs) have harnessed their formidable representational capabilities to attain remarkable performance in image retrieval models. Nonetheless, in cases where malicious actors introduce adversarial perturbations into the test dataset, the retrieval model may readily yield results that are either irrelevant or intentionally manipulated by the attacker. Specifically, the targeted attack is notable for producing predefined results, thereby inflicting a more adverse impact on retrieval performance. While adversarial purification has demonstrated effectiveness in countering adversarial attacks, its application in retrieval tasks remains unexplored. Addressing these concerns, we introduce a free-trained purification model denoted as **RetPur** aimed at purifying adversarial test dataset, thereby mitigating the issue of targeted attacks within both uni-modal and cross-modal retrieval systems. **RetPur** employs a pre-trained diffusion model, offering a plug-and-play convenience, while utilizing adversarial samples as conditioning factors to guide image generation, thereby enhancing task accuracy. In terms of retrieval system architecture, our study pioneers the incorporation of adversarial purification tasks into uni-modal (Image-to-Image) and cross-modal (Image-to-Image, Image-to-Text) hash retrieval systems, specifically tailored to image retrieval scenarios. Furthermore, we explore the application of adversarial purification tasks to a wider array of attacks, including both generative and iterative approaches. Through an extensive series of experiments, it can be concluded that the purified dataset exhibits retrieval performance in the retrieval systems that is closely akin to that of the original dataset, even across different attacks and modalities.

## 1 Introduction

Deep Neural Networks (DNNs) perform well in image processing tasks such as image classification, image retrieval, semantic segmentation, and object localization. However, extensive research has shown that trained DNNs are prone to interference from adversarial samples (McDaniel et al. (2016);Papernot et al. (2016);Ma et al. (2019);Cohen et al. (2020);Mayer & Timofte (2020);Wu et al. (2021)) and output incorrect task results. Among them, adversarial samples refer to the attacker exerting small perturbations on the test set data, which makes it difficult to distinguish it from original data with the naked eye, but can cause the model to misjudge.

Researchers have conducted extensive research on defense against adversarial attacks. These defense methods are mainly divided into three types: adversarial training (Sarkar et al. (2021);Deng et al. (2021);Zhao et al. (2022)), adversarial detection (Liu et al. (2018);Yang et al. (2021)), and adversarial purification (Nie et al. (2022);Wu et al. (2023);Kim et al. (2022)). The purpose of adversarial training is to involve adversarial instances or optimize instances in the worst-case scenario during the training process of neural networks. Although it improves the robustness of the target classifier against adversarial attacks of visible types, it often cannot handle adversarial attacks or image damage of invisible types. Adversarial detection (Zhang et al. (2022)) aims to determine whether a given sample is an adversarial one based on the discrepancy between natural and adversarial distributions. However, it is difficult to quantify the threshold for distinguishing adversarial samples from original data, and it is prone to errors in determining the distribution of the data. Unlike the first two methods, adversarial purification (Nie et al. (2022);Wang et al. (2022)) only applies to the test set data and has a plug-and-play feature, without increasing the complexity of the model. Specifically, adversarial purification methods typically use learning to generate models as purifica-

tion models, converting the attacked test set data into original data and inputting it into the attacked model.

Currently, many works about adversarial examples have been studied in image classification, but very few researches focus on the security of deep hashing based retrieval models. Different from the typical classification, hash retrieval models (Li et al. (2015);Cao et al. (2018);;Bai et al. (2020a);Yu et al. (2021)) aims to learn the semantic similarity between images, and its final outputs are discrete binary codes instead of categories. Especially target attacks can cause query results to return fixed categories of information, completely interfering with the performance of the retrieval system, as shown in the Appendix A.

This paper is first to consider applying adversarial purification tasks to the field of image retrieval. Inspired by recent works (Nie et al. (2022);Wang et al. (2022)), we propose a diffusion-based purification model **RetPur** to purify the attacked retrieval dataset. The underlying principle of the model can be outlined as follows: Attackers obtain retrieval datasets and proceed to launch attacks by introducing interference. Adversarial datasets are input into the **RetPur** model, wherein a forward denoising process is applied to generate pristine noise-free images. This process iteratively eliminates noise, guided by the input adversarial samples, ultimately resulting in the production of purified datasets. The proposed model has plug-and-play convenience, and the purified test dataset demonstrates superior anti-attack and retrieval performance on the hash retrieval model.

We conducted extensive experiments to evaluate the robustness of our model in both uni-modal image retrieval tasks and cross-modal image retrieval systems. Owing to the heightened destructive potential of targeted attacks, our choice for the unimodal retrieval task encompassed target generative attacks and target iterative attacks (Bai et al. (2020b)). For cross-modal retrieval tasks, we exclusively employed TA-DCH (Wang et al. (2023)) as the attack method, as it is specifically designed for cross-modal models. In order to assess the efficacy of the purification model, we curated original retrieval datasets, purified original retrieval datasets, and purified adversarial datasets. Subsequently, we conducted a comparative analysis of their retrieval outcomes within the retrieval model, utilizing MAP and T-MAP values as performance metrics. Based on experimental findings, it can be concluded that the retrieval outcomes of both the purified original datasets and the purified adversarial datasets within the image retrieval model closely approximate those of the original original datasets, with a MAP value error of less than $2\%$. This demonstrates the versatility of the proposed **RetPur** model, which can be effectively employed across a broader spectrum of purification tasks.

## 2 RELATED WORK

### 2.1 ADVERSARIAL ATTACKS AGAINST HASHING RETRIEVAL

At present, there are few studies on adversarial attacks against retrieval systems. Similar to image classification models, DNN-based image retrieval models are also susceptible to adversarial sample attacks. Attacks against image retrieval tasks belong to black box attacks, which are divided into two types: target black box attacks and non target black box attacks. According to different retrieval scenarios, the retrieval model is divided into uni-modal retrieval model and cross-modal retrieval model. Existing adversarial attacks against retrieval systems include non-targeted (Yang et al. (2018);Xiao et al. (2020);Feng et al. (2020);Li et al. (2021b)) and targeted attacks (Bai et al. (2020b);Xiao & Wang (2021);Hu et al. (2021);Wang et al. (2021b)) against uni-modal retrieval, non-targeted attacks (Li et al. (2019);Li et al. (2020);Li et al. (2021a);Zhu et al. (2023);Zhang et al. (2023)) against cross-modal retrieval, and targeted attacks (Wang et al. (2023)) against cross-modal retrieval. According to the different ways of generating adversarial samples, the above attacks can be divided into iterative attacks and generative attacks.

**Targeted attacks against hashing retrieval.** Non-targeted attacks aim to make retrieval systems return results irrelevant to the query. While targeted ones not only realize the above purpose but also disguise adversarial examples into the attacker's specified category, such as violence, bloodiness, and pornography. Compared with non-targeted attacks, targeted attacks cause retrieval systems to provide users with insufferable malicious information, which seriously reduces their trustworthiness. Based on this, we select target attacks as the research focus and considers two retrieval tasks: uni-modal tasks and cross-modal tasks. At the same time, we also analyzed the impact of the proposed algorithm on iterative and generative attacks.

## 2.2 DEFENSE STRATEGIES AGAINST TARGETED ATTACKS

For deep hashing-based retrieval, Wang et al. (2021a) proposed the first effective adversarial training algorithm based on the targeted attack (dubbed ATRDH here) by narrowing the semantic gap between the adversarial samples and the original samples in the Hamming space. Adversarial training (Goodfellow et al. (2014b), Madry et al. (2017)) aims to augment the training data with generated adversarial examples, which is the most common training strategy against various adversarial attacks. However, models trained by adversarial training can only defend against specific attack methods used during training, and cannot resist attacks from other attack methods, resulting in poor generalization performance.

**Adversarial purification.** Refer to Appendix B, a generative model that can restore original images from attacked images is trained and used as a preprocessor in adversarial purification, which is more convenient than adversarial training. Samangouei et al. (2018) proposed defense-GAN, a generator that can restore original images from attacked images. Song et al. (2017) revealed how to detect and purify adversarial samples using an autoregressive generative model. Srinivasan et al. (2021) presented a purification approach for denoising autoencoders using the Metropolis Adjusted Langevin algorithm (MALA). Grathwohl et al. (2019), Du & Mordatch (2019) and Hill et al. (2020) proposed and demonstrated that MCMC wich EBMs can purify adversarial examples. Similarly, Yoon et al. (2021) used the denoising score-based model for purification. Due to the development of diffusion models in the field of image generation, some works (Nie et al. (2022);Wang et al. (2022);Wu et al. (2022)) consider using the DDPM model and its improved versions as purification models to purify disturbed test set data, but only consider using simple datasets such as cifar10 and imagenet for classification tasks. Recently, some studies have considered classification or segmentation tasks for more complex datasets, such as 3D point clouds (Sun et al. (2022)), self supervised vascular segmentation (Kim et al. (2022)), and audio (Wu et al. (2023)). Based on the above research foundation, we propose to apply adversarial purification methods to the field of hash retrieval.

## 3 RETPUR: DIFFUSION-BASED PURIFICATION MODEL

### 3.1 STRUCTURE OF THE GUIDED DIFFUSION MODEL

We propose to use a pre-trained DDPM to purify adversarial samples, which is one of the most widely used diffusion model (Song & Ermon (2019);Song et al. (2020a);Song et al. (2020b)). The DDPM model consists of two processes: the forward diffusion process and the reverse generation process. Briefly, the DDPM model is a process of adding and removing noise.

In the DDPM model, both of the two processes are defined by Markov chains with a length of $T$. Assume $p_{data}$ is the distribution of all images $x$ in the dataset, $p_\theta$ is the distribution of generate images. It is worth noting that $p_\theta$ tends to approach $p_{data}$, i.e. $p_\theta \sim p_{data}$.

**The forward diffusion process.** Formally, let $x^0 \sim p_{data}$, the diffusion step $t$ is denoted by a superscript. Set $p_{noise} = N(0_{3N}, I_{3N \times 3N})$ as the result of the diffusion process, and $N$ is the Gaussian distribution. The diffusion process from original data $x^0$ to $x^T$ is defined as below:

$$q(x^1, \cdots, x^T | x^0) = \prod_{t=1}^{T} q(x^t | x^{t-1}), \tag{1}$$

$$q(x^t | x^{t-1}) = N(x^t; \sqrt{1 - \beta_t} x^{t-1}, \beta_t I), \tag{2}$$

where $q$ is the distribution of original data in the forward diffusion process, $q(x^T)$ is close to $p_{noise}$. $\beta_t$ are the diffusion coefficients and their values are predefined small positive constants.

According to Ho et al. (2020), there is a closed form expression for $q(x^t | x^0)$. Define constants $\alpha_t = 1 - \beta_t, \bar{\alpha}_t = \prod_{i=1}^{t} \alpha_i$. Then, we have $q(x^t | x^0) = N(x^t; \sqrt{\alpha_t} x^{t-1}, (1 - \alpha_t)I)$. Therefore, when $T$ is large enough, $\bar{\alpha}_t$ goes to 0, and $q(x^T | x^0)$ becomes close to the final distribution $p_{noise}$. Note that $x^t$ can be directly sampled through the following equation:

$$x^t = \sqrt{\bar{\alpha}_t} x^0 + \sqrt{1 - \bar{\alpha}_t} \epsilon, \text{ where } \epsilon \text{ is a standard Gaussion noise.} \tag{3}$$

**The reverse sampling process.** Let $x^T \sim p_{noise}$ be a latent variable. It is a Markov process that predicts and eliminates the noise added in the diffusion process. The reverse process from latent $x^T$

to purified image $x^0$ is defined as:

$$p_\theta(x^0, \cdots, x^{T-1}|x^T) = \prod_{t=1}^{T} p_\theta(x^{t-1}|x^t), \tag{4}$$

$$p_\theta(x^{t-1}|x^t) = N(x^{t-1}; \mu_\theta(x^t, t), \sigma_t^2 I), \tag{5}$$

where $\mu_\theta(x^t, t)$ is a neural network parameterized by $\theta$, and the variance $\sigma_t^2$'s can be either time-step dependent constants (Ho et al. (2020)) or learned by a neural network (Nichol & Dhariwal (2021)). To generate a sample, we first sample $x^T \sim p_{noise}$, then draw $x^{t-1} \sim p_\theta(x^{t-1}|x^t)$ for $t = T, T-1, \cdots, 1$, and finally outputs purified data $x^0$.

**Conditional gradient.** The work (Wang et al. (2022)) proposed a novel Guided DDPM, in which the authors use the adversarial images $x_{adv}$ to guide the reverse process of the DDPM. This method is used to purify classified datasets, including CIFAR10 and ImageNet. Experiments have shown that the purified images performs better in classification tasks than DDPM. The difference from original DDPM is that the gradient descent in formula 5 is regulated by a conditional gradient.

$$p_\theta(x^{t-1}|x^t, x_{adv}) = N(x^{t-1}; \mu_\theta(x^t, t) - s\sigma_t^2 \nabla_{x^t} D(x^t, x_{adv}^t), \sigma_t^2 I), \tag{6}$$

where $s$ represents the guidance scale, $-\sigma_t^2 \nabla_{x^t} D(x^t, x_{adv}^t)$ is the conditional gradient. The mathematical derivation of the conditional gradient will be presented in the Appendix C.

## 3.2 THE BASIC IDEA OF HASH RETRIEVAL PURIFICATION MODEL

According to previous research work, this imperceptible disturbance can be divided into two situations: One is a disturbance generation method based on optimization and heuristic algorithms, as shown in formula 7. The other is a generative disturbance generation method, as shown in formula 8. Assuming $x$ is a original image and $x_{adv}$ is an adversarial image. $x_{adv(1)}$ and $x_{adv(2)}$ represent adversarial samples generated by two different generation methods, respectively. The formula is expressed as follows.

$$x_{adv(1)} = x + \delta, \tag{7}$$

where $\delta$ represents noise that does not affect the structural information of the original image and can be seen as redundant information of the image.

$$x_{adv(2)} = Generator(x + Decoder(s)), \tag{8}$$

where $s$ represents the target semantics generated from the target attack label. The attacker generates an image of the same size as a original image based on the target semantics, concatenates it after the original image, and then inputs the synthesized image into the generator to obtain adversarial samples.

Due to the structural characteristics of the DDPM model itself, Gaussian noise is added to submerge and counteract interference, while removing the interference of Gaussian noise. As shown in Figure 1, we propose the following model:

It is worth noting that we considers two cases of $x_{adv}$, namely it has two forms: $x_{adv} = x_{adv(1)}$ or $x_{adv} = x_{adv(2)}$. Given an adversarial example $x_{adv}$ as the input at $t = 0$, i.e., $x^0 = x_{adv}$. We first diffuse the adversarial image $x_{adv}$ for $T$ steps following Equation 3:

$$x^t = \sqrt{\bar{\alpha}_t} x_{adv} + \sqrt{1 - \bar{\alpha}_t}\epsilon, \ \epsilon \sim N(0, I). \tag{9}$$

Next, we condition the reverse denoising process of the DDPM on the adversarial image $x_{adv}$. Specifically, we adapt the reverse denoising distribution $p_\theta(x^{t-1}|x^t)$ in Equation 5 to a conditional distribution $p_\theta(x^{t-1}|x^t, x_{adv})$. Let noise image $x^T$ as input and U-Net model as a model for fitting parameter $\mu_\theta$ and $\epsilon_\theta$. The representation of reverse generated image $x^0$ can be obtained by using Equation 6.

$$x^0 = \textbf{U-Net}(x^T, \mu_\theta, \epsilon_\theta, D(x^t, x_{adv}^t), s, T), \ t = T, \ldots, 1, \tag{10}$$

where $x_{adv}^t$ is the guidance at step $t$. It's obtained by diffusing $x_{adv}$ $t$ steps according to Equation 9.

Combining the Representation of Equation 9 and Equation 10, the whole purification operation can be denoted as a function **RetPur**: $\mathbb{R}^d \times \mathbb{R} \to \mathbb{R}^d$:

$$\textbf{RetPur}(x_{adv}) = \textbf{U-Net}(\sqrt{\bar{\alpha}_T}x_{adv} + \sqrt{1 - \bar{\alpha}_T}\epsilon, \mu_\theta, \epsilon_\theta, D(x^t, x_{adv}^t), s, T), \ t = T, \ldots, 1. \tag{11}$$

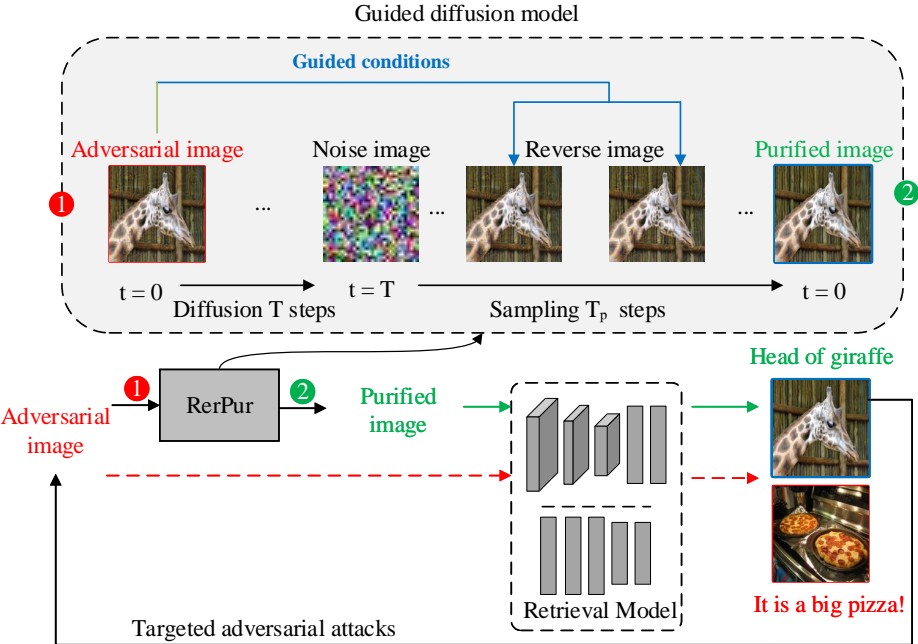

Figure 1: An illustration of **RetPur**. Given a pre-trained diffusion model with $T$ diffusion steps, we incrementally introduce minor noise to the adversarial image during the forward diffusion process until it transforms into a pure noise image. Following this, in the purification step denoted as $T_p$, we employ the input adversarial image as a guiding condition and gradually eliminate noise using pre-trained model parameters, resulting in the acquisition of purified images.

### 3.3 LEARN ALGORITHM

As shown in Algorithm 1 and Algorithm 2, we divide the learning algorithm of the purification model into two parts: training and sampling. For simplicity, this paper ignores the fact that the forward process variances $\beta_t$ are learnable by reparameterization and instead fix them to constants. Thus, in our implementation, the approximate posterior $q$ has no learnable parameters, we just need to discuss our choices in $p(x^{t-1}|x^t, x_{adv})$.

---

**Algorithm 1** Training

**Input:** $x^0 \sim p_{data}$, diffusion step $T$, initialize a U-net as $\epsilon_\theta(x^0, 0)$, learning rate $\eta$
**Output:** noise predictor $\epsilon_\theta(x^t, t)$
1: **repeat**
2:    $x^0 \sim p_{data}$
3:    $t \sim Uniform(\{1, \ldots, T\})$
4:    $\epsilon \sim N(0, I)$
5:    Take gradient descent step on:
     $\nabla_\theta \| \epsilon - \epsilon_\theta(\sqrt{\bar{a}_t} x^0 + \sqrt{1 - \bar{\alpha}_t}\epsilon, t) \|$
6:    update $\epsilon_\theta(\cdot)$:
     $\epsilon_\theta(x^t, t) = \epsilon_\theta(x^{t-1}, t-1) - \eta \cdot \nabla_\theta(\cdot)$
7: **until** converged
8: **return** $\epsilon_\theta(x^t, t)$

**Algorithm 2** Sampling

**Input:** $x^T \sim N(0, I)$, purify step $T_p$, distance metric $D(\cdot)$, guidance scale $s$
**Output:** $x^0$
1: **for** $t = T_{pur}, \ldots, 1$ **do**
2:    $z \sim N(0, I)$ if $t > 1$, else $z = 0$
3:    compute conditional gradient:
     $\nabla G = -s\nabla_{x^t} D(x^t, x_{adv}^t)$
4:    $\mu_\theta(x^{t-1}, t-1)$
     $= \frac{1}{\sqrt{\bar{\alpha}_t}}(x^t - \frac{\beta_t}{\sqrt{1-\bar{\alpha}_t}}\epsilon_\theta(x^t, t)) + \nabla G \cdot$
     $\epsilon_\theta(x^t, t))$
5:    $x^{t-1} = \mu_\theta(x^{t-1}, t-1) + \sigma z$
6: **end for**
7: **return** $x^0$

---

**Training.** Similar to Ho et al. (2020), we use $\mu_\theta$ and $\epsilon_\theta$ as parameters predicted by the U-Net model to fit the distribution of original data.

$$\mu_\theta(x^{t-1}, t-1) = \frac{1}{\sqrt{\bar{\alpha}_t}}(x^t - \frac{\beta_t}{\sqrt{1 - \bar{\alpha}_t}}\epsilon_\theta(x^t, t)), \tag{12}$$

where $\epsilon_\theta$ is a function approximator intended to predict $\epsilon$ from $x^t$. To sample $x^{t-1} \sim p_\theta(x^{t-1}|x^t)$ is to compute $x^{t-1} = \frac{1}{\sqrt{\bar{\alpha}_t}}(x^t - \frac{\beta_t}{\sqrt{1-\bar{\alpha}_t}}\epsilon_\theta(x^t,t)) + \sigma z$, where $z \sim N(0,I)$. It can be seen that the only parameter that needs to be estimated is $\epsilon_\theta$.

Ho et al. (2020) found it beneficial to sample quality to train on the following variant of the variational bound:

$$L_{simple}(\theta) := \mathbb{E}_{t,x^0,\epsilon}[\|\epsilon - \epsilon_\theta(\sqrt{\bar{\alpha}_t}x^0 + \sqrt{1-\bar{\alpha}_t}\epsilon, t)\|^2], \tag{13}$$

where $t$ is uniform between $1$ and $T$.

By adopting the idea of Equation 13, we propose a loss function for the image purification operation performed by the **RetPur**:

$$L_{purify}(\theta) := \mathbb{E}_{t,x_{adv},\epsilon}[\|\epsilon - \epsilon_\theta(\sqrt{\bar{\alpha}_T}x_{adv} + \sqrt{1-\bar{\alpha}_T}\epsilon, t)\|^2], \tag{14}$$

where $x_{adv}$ has two forms. During the model training process, we only need to train the noise predictor $\epsilon_\theta$. And the training method of the noise predictor is shown in Algorithm 1.

**Sampling.** After the training process is completed, we can obtain the trained noise predictor $\epsilon_\theta(x^t, t)$ and use it to update parameters, gradually sampling and generating images. The update method for parameter $\mu_\theta$ are as follows:

$$\mu_\theta(x^{t-1}, t-1) = \frac{1}{\sqrt{\bar{\alpha}_t}}(x^t - \frac{\beta_t}{\sqrt{1-\bar{\alpha}_t}}\epsilon_\theta(x^t,t)) - s\nabla_{x^t}D(x^t, x_{adv}^t)\epsilon_\theta(x^t,t), \tag{15}$$

where $s$ is a constant which represents gradient scale, $D(\cdot)$ represents some distance metric. In our experiments, we adopt the mean square error (MSE) or negative structure similarity index measure (SSIM) as our distance metric $D(\cdot)$. Thus, we can get the update formula of $x^{t-1}$:

$$x^{t-1} = \frac{1}{\sqrt{\bar{\alpha}_t}}(x^t - \frac{\beta_t}{\sqrt{1-\bar{\alpha}_t}}\epsilon_\theta(x^t,t)) - s\nabla_{x^t}D(x^t, x_{adv}^t)\epsilon_\theta(x^t,t) + \sigma z, \; z \sim N(0,I). \tag{16}$$

## 4 EXPERIMENTS

### 4.1 EXPERIMENTAL SETTINGS

We conducted two sets of experiments on the uni-modal hash retrieval models and the cross-modal hash retrieval models.

**Dataset.** We conduct extensive experiments on five widely used retrieval datasets: Wikipedia (Pereira et al. (2013)), IAPR-TC12 (Escalante et al. (2010)), FLICKR-25K (Huiskes & Lew (2008)), MS-COCO (Lin et al. (2014)) and NUS-WIDE (Chua et al. (2009)). The division of datasets differs between uni-modal and cross-modal hash retrieval systems, as detailed in the Appendix D.

**Attacked models and Attack methods.** For the uni-modal experiment, we select representative retrieval systems DPSH and DSH, and introduce two types of attacks: iterative attacks exemplified by P2P and DHTA, along with generative attacks represented by ProS-GAN. For the cross-modal experiment, we opt for representative retrieval systems DGCPN and DADH, and incorporate the target attack TA-DCH. The specific settings are detailed in Appendix E.

**Evaluation Metrics.** Following the previous works (Jiang & Li (2017), Bai et al. (2020b)), we use MAP and T-MAP to evaluate the performance of Purification model. MAP is a widely used metric to evaluate the performance of cross-modal retrieval. T-MAP is slightly different from MAP in that the original labels of the query set are replaced with the target labels specified by the attacker. Since T-MAP uses the target labels as the test labels, the higher the T-MAP, the stronger the targeted attack.

**Implementation details:** In experiments, our proposed model is implemented via PyTorch and a NVIDIA 3090 GPU. Training attack methods and retrieval models on the same datasets. All attack methods have a training bit of 32. The iterative attack methods DHTA and P2P have an iteration count of 2000 and 500, while the iterations of the generative attack methods ProS-GAN and TA-DCH are 100 and 150, respectively. For the pre-trained diffusion model, we use models from Ho et al. (2020) and Dhariwal & Nichol (2021). Diffusion step $T$ is set to 1,000 and purify step $T_p$ is set to 45. In the calculation of conditional gradients, SSIM is selected as the default distance metric function, with a guidance scale $s$ of 1,000. We adopt the Adam with learning rate $10^{-4}$ to optimize the proposed **RetPur** model.

Table 1: Quantitative results (%) of P2P, DHTA, ProS-GAN (Bai et al. (2020b);Wang et al. (2021a)) attacks targeting Image to Image (I2I) retrieval tasks. Original uses benign images to retrieve images, Attack methods use generated adversarial samples to retrieve images, and **RetPur** purifies the adversarial samples generated by attack methods and generates purified samples to retrieve images.

| I2I | | FLICKR-25K | | MS-COCO | | NUS-WIDE | |
|---|---|---|---|---|---|---|---|
| | | DPSH | DPH | DPSH | DPH | DPSH | DPH |
| MAP | original | 81.06 | 77.07 | 63.69 | 58.95 | 73.06 | 69.92 |
| | P2P | 65.81 | 62.51 | 52.52 | 44.61 | 61.97 | 54.81 |
| | **RetPur-P2P** | **80.17** | **76.58** | **63.15** | **58.56** | **72.62** | **69.89** |
| | DHTA | 61.09 | 61.03 | 46.17 | 42.43 | 51.98 | 53.89 |
| | **Retpur-DHTA** | **80.14** | **76.64** | **63.14** | **58.57** | **72.72** | **69.83** |
| | ProS-GAN | 63.29 | 64.58 | 45.97 | 44.54 | 53.45 | 50.73 |
| | **RetPur-ProS** | **79.50** | **75.99** | **62.16** | **57.76** | **71.56** | **69.14** |
| T-MAP | original | 73.67 | 73.21 | 50.53 | 49.22 | 59.17 | 60.27 |
| | P2P | 82.04 | 80.51 | 48.69 | 53.17 | 67.81 | 71.28 |
| | **RetPur-P2P** | **63.60** | **64.06** | **43.21** | **42.73** | **57.06** | **57.01** |
| | DHTA | 88.39 | 83.85 | 60.01 | 59.67 | 69.68 | 75.68 |
| | **Retpur-DHTA** | **63.95** | **64.28** | **43.48** | **42.91** | **57.44** | **57.21** |
| | ProS-GAN | 92.76 | 86.02 | 72.36 | 65.25 | 90.00 | 76.31 |
| | **RetPur-ProS** | **72.60** | **72.31** | **46.49** | **46.41** | **54.25** | **53.69** |

## 4.2 MAIN RESULTS

### 4.2.1 UNI-MODAL HASH RETRIEVAL PURIFICATION

Referring to the experiment conducted on the three attacks in work (Wang et al. (2021b)), we selected the same three datasets: FLICKR-25K, MS-COCO, NUS-WIDE.

**Purification against iterative attacks:** Regarding the method of generating adversarial samples, we select two gradient based iterative attack methods as the baseline, including P2P and DHTA (Bai et al. (2020b)). As shown in the Table 1, the purified image retrieval results approximate the retrieval results of the original image. In addition, compared to the results of using adversarial sample retrieval, the MAP value significantly increased and the T-MAP value significantly decreased. Wang et al. (2021a) considers using adversarial training methods on both FLICKR-25K and NUS-WIDE datasets to improve the performance of DPH retrieval model. For the FLICKR-25K dataset, under P2P and DHTA attacks, the T-MAP values of the purified data on the DPH model were $64.06$ and $64.28$, respectively, which were lower than the retrieval results of the adversarial samples on the DPH model after adversarial training (Wang et al. (2021a)). For the NUS-WIDE dataset, the retrieval results of the two defense methods on the two attacks are very similar, with a difference of within $0.4\%$. This confirms that our proposed purification model performs better than existing adversarial training methods.

**Purification against generative attacks:** We use ProS-GAN (Wang et al. (2021b)) as a generative attack, which is the first target generative attack based on deep hash retrieval systems. Instead of heuristic selection of a hash code as a representative of the target label, a flexible prototype code is generated in PrototypeNet using semantic invariance as the expected pillar of the target label in the optimization view. Therefore, compared to iterative attacks, ProS-GAN has shorter training time and stronger attack ability. As shown in Table 1, our purification model performs excellently. Compared to the MAP values of the adversarial samples generated by ProS-GAN, the purified data showed a $10\%$ to $20\%$ increase on the retrieval model, with a difference of less than $1.6\%$ compared to the original data. It is worth noting that the T-MAP values of the purified data on all datasets are smaller than the T-MAP values retrieved from the original data, which proves that the performance of purified data retrieval without being classified into target categories is better than that of original data. It is concluded that the adversarial purification model can effectively resist uni-modal target generative attacks.

Table 2: Quantitative results (%) of TA-DCH (Wang et al. (2023)) attacks targeting Image to Image (I2I) retrieval tasks. Original uses benign images to retrieve images, TA-DCH uses generated adversarial samples to retrieve images, and **RetPur** purifies the adversarial samples generated by TA-DCH and generates purified samples to retrieve images.

| I2I | | WIKI | | IAPR | | FLICKR-25K | | MS-COCO | | NUS-WIDE | |
|---|---|---|---|---|---|---|---|---|---|---|---|
| | | DGCPN | DADH | DGCPN | DADH | DGCPN | DADH | DGCPN | DADH | DGCPN | DADH |
| MAP | original | 38.09 | 44.93 | 64.74 | 69.35 | 81.99 | 80.58 | 61.92 | 58.49 | 77.91 | 75.60 |
| | TA-DCH | 28.63 | 39.80 | 39.28 | 38.77 | 67.05 | 65.30 | 44.94 | 55.15 | 43.27 | 46.64 |
| | **RetPur** | **36.22** | **42.93** | **63.20** | **65.70** | **80.56** | **83.21** | **59.60** | **56.46** | **76.57** | **73.71** |
| T-MAP | original | 13.33 | 12.01 | 31.17 | 33.43 | 56.71 | 57.07 | 34.27 | 36.91 | 35.45 | 35.92 |
| | TA-DCH | 42.37 | 23.80 | 52.29 | 79.85 | 88.57 | 96.27 | 61.17 | 45.83 | 87.31 | 86.48 |
| | **Retpur** | **18.66** | **18.44** | **33.61** | **35.74** | **59.25** | **58.64** | **39.90** | **37.96** | **38.34** | **40.58** |

Table 3: Quantitative results (%) of TA-DCH (Wang et al. (2023)) attacks targeting Image to Text (I2T) retrieval tasks. Original uses benign images to retrieve texts, TA-DCH uses generated adversarial samples to retrieve texts, and **RetPur** purifies the adversarial samples generated by TA-DCH and generates purified samples to retrieve texts.

| I2T | | WIKI | | IAPR | | FLICKR-25K | | MS-COCO | | NUS-WIDE | |
|---|---|---|---|---|---|---|---|---|---|---|---|
| | | DGCPN | DADH | DGCPN | DADH | DGCPN | DADH | DGCPN | DADH | DGCPN | DADH |
| MAP | original | 37.79 | 43.15 | 65.24 | 72.49 | 80.56 | 78.45 | 65.21 | 62.28 | 76.42 | 77.65 |
| | TA-DCH | 28.30 | 37.94 | 38.94 | 39.10 | 65.36 | 62.76 | 45.85 | 58.11 | 43.17 | 44.17 |
| | **RetPur** | **36.15** | **41.35** | **63.49** | **69.27** | **79.10** | **84.87** | **63.63** | **60.36** | **75.38** | **75.21** |
| T-MAP | original | 13.34 | 12.11 | 31.37 | 37.60 | 56.54 | 62.39 | 34.25 | 38.67 | 35.64 | 39.23 |
| | TA-DCH | 42.34 | 23.67 | 52.07 | 81.11 | 85.07 | 93.98 | 62.82 | 47.06 | 87.16 | 84.34 |
| | **RetPur** | **18.65** | **18.61** | **33.76** | **40.69** | **59.19** | **62.89** | **41.10** | **39.28** | **38.23** | **42.97** |

### 4.2.2 CROSS-MODAL HASH RETRIEVAL PURIFICATION

Considering the wider applicability of the cross-modal retrieval model, we selected the adversarial samples generated by the TA-DCH (Wang et al. (2023)) attack as input for the purification model to explore whether the purification task is suitable for querying multimodal data. Specifically, we consider the tasks of Image to Image and Image to Text. We use the **RetPur** model to purify the adversarial retrieval dataset, input the purified retrieval set into a cross modal hash retrieval system, and retrieve semantically similar text and images in the database. As shown in Table 2 and Table 3, The purification model we proposed is also suitable for cross-modal retrieval tasks. The purified image performs well in both image to image and image to text tasks, far exceeding the results of the attack and approaching the results of the original image query. Using the MAP value as an indicator, the performance of the purified data in the retrieval model has significantly improved compared to the adversarial samples, with a performance gap of less than 2% compared to the original data. Among them, in the experiment using the DADH retrieval model on the FLACKR-25K dataset, our method outperformed the original data by at least 3% in both Image to Image and Image to Text retrieval tasks. When T-MAP is used as an evaluation indicator, our method's results on the retrieval model are close to those of the original data, and far lower than the results of adversarial samples retrieval. It is concluded that the adversarial purification model can effectively resist cross-modal target generative attacks.

### 4.2.3 PURIFY ORIGINAL DATASETS

Considering more complex scenarios, attackers may randomly attack the retrieved dataset. In this case, some of the data in the dataset after the attack is original data, while others are adversarial samples. Therefore, more general purification tasks need to be considered. The performance of the purification model needs to ensure that the retrieval effect after original data purification is similar to that after adversarial sample purification, both approaching the retrieval effect of original effect. We designed experiments to purify original datasets on both uni-modal and cross-modal retrieval tasks,

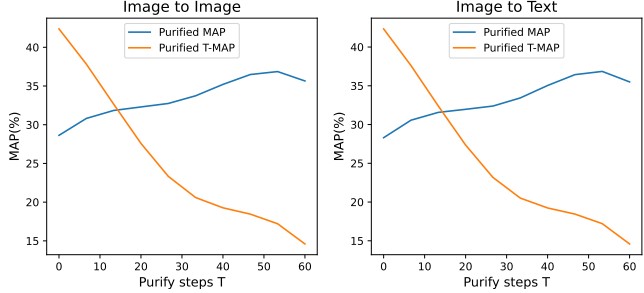

Figure 2: The correlation analysis of purification performance and purify step on Image to Image and Image to Text tasks.

Table 4: Purify original datasets of uni-modal retrieval task.

|  | Image to Image | | Image to Text | |
|---|---|---|---|---|
|  | MAP | T-MAP | MAP | T-MAP |
| SSIM | 36.22 | 18.66 | 36.15 | 18.65 |
| MSE | 36.52 | 14.52 | 36.38 | 14.39 |

and then compared the purified retrieval results with the original image retrieval results to analyze the impact of purification tasks on the performance of original data retrieval. As shown in the Table 7, Table 8 and Table 9 of Appendix F, the results of uni-modal tasks show that the retrieval results of the purified dataset is only slightly reduced compared to the original dataset, within $2\%$. Based on the conclusions of previous experiments, the difference between the retrieval results of the purified adversarial samples and the original data is within $2\%$. A more general conclusion can be drawn that the difference in retrieval performance between the purified dataset and the original dataset is within $2\%$. Among them, the dataset contains both original data and adversarial samples.

### 4.3 ABLATION STUDIES

This section explores the impact of different parameter settings on the performance of the purification model **RetPur**. We set the experimental dataset as the WIKI dataset, the attack method is TA-DCH attack and the retrieval model being attacked is DGCPN.

**Purify step** $T_{pur}$**.** As shown in Figure 2, there is a game process between purification performance and visual perceptibility. With the increase of purify step, the performance of the **RetPur** model for adversarial examples increases rapidly.

**Distance Metric.** The guidance conditions $\nabla x^t D(x^t, x^t_{adv})$ are the gradient value of the distance function between the generated samples and the adversarial samples. Consider the impact of different distance metric $D(\cdot)$ on conditional generation, we compared two different distance functions, SSIM and MSE. As shown in Table 4, Compared to using SSIM, using MSE as a conditional gradient guide, the purified data has slightly higher MAP values and T-MAP values on the hash retrieval model.

## 5 CONCLUSION

In this paper, we propose a diffusion-based purification model to defend against target black-box attacks in hash retrieval systems. To assess the effectiveness of **RerPur**, we evaluate our approach in both uni-modal retrieval tasks and cross-modal retrieval tasks, considering two types of attacks: iterative attacks and generative attacks. The experiments demonstrate that the purified adversarial samples and purified original data yield retrieval model results that are close to those of the original data, with significant improvements compared to the results of the attacked retrieval. Furthermore, the **PerPur** model outperforms existing adversarial training method ATRDH (Wang et al. (2021a)). Therefore, the model we propose can be effectively applied in the field of hash retrieval.

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

# APPENDIX

## A EXAMPLE OF TARGET RETRIEVAL ATTACK

As shown in Figure 3, when the aircraft image is input into the retrieval system, the query text and images returned depict a panda. In the case of targeted attacks, inputting images with any semantic content into the retrieval system results in specific categories and corresponding text being returned.

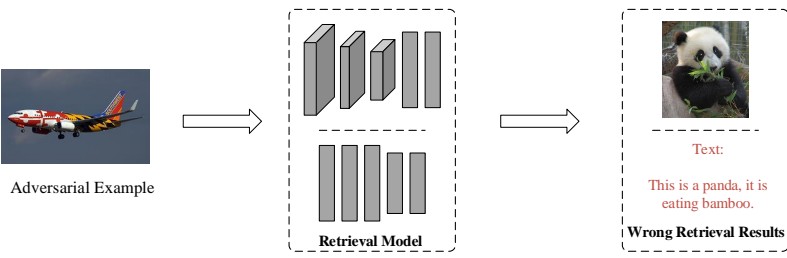

Figure 3: Wrong retrieval results after attack. Inputting aircraft images with adversarial interference into the retrieval system returns both textual and visual results depicting a panda.

## B DEFINITION OF PREPOSSESSING METHODS

Essentially, the adversarial purification is one of the prepossessing methods. Preprocessing, in which input images are preprocessed using auxiliary transformations before classification, is another strategy for adversarial defense. Let $f_\theta$ be a preprocessor for image transformation. The goal of the training is then:

$$\min_{\phi,\theta} \mathbb{E}_{p_{data}(x,y)}[max_{x'\in B(x)}L(g_\theta(f_\theta(x')),y)]$$

An average over recognized types of adversarial attacks or an average over stochastically altered inputs can be used to approximate the maximum over the threat model $B(x)$. Adding stochasticity to the input images, as well as discontinuous or non-differentiable transforms (Guo et al. (2017);Dhillon et al. (2018);Buckman et al. (2018);Xiao et al. (2019)), makes gradient estimation concerning the loss function $\nabla_x L(g_\theta(f_\theta(x')),y)$ harder to attack.

## C PROOF OF CONDITIONAL GRADIENT

Refer to the work (Wang et al. (2022)), we provide a proof of conditional gradient in the formula 6. By using adversarial image $x_{adv}$ as a condition for the reverse denoising process of the DDPM model, we can obtain a distribution $p_\theta(x^{t-1}|x^t, x_{adv})$. It is a conditional distribution of denoising distribution $p_\theta(x^{t-1}|x^t)$ in the formula 5. Dhariwal & Nichol (2021) proved that

$$\log p_\theta(x^{t-1}|x^t, x_{adv}) = \log[p_\theta(x^{t-1}|x^t)p(x_{adv}|x^t)] + C_1 \approx \log p(z) + C_2, \quad (17)$$

where distribution $z \sim N(z; \mu_\theta(x^t, t) + \sigma_t^2 g, \sigma_t^2 I)$, $g = \nabla_{x^t} \log p(x_{adv}|x^t)$, $C_1$ and $C_2$ are some constants. Due to the original distribution $p_\theta(x^{t-1}|x^t) \sim N(\mu_\theta(x^t, t), \sigma_t^2 I)$, $\sigma_t^2 g$ is the conditional gradient of the required solution.

$$\sigma_t^2 g = \sigma_t^2 \nabla_{x^t} \log p(x_{adv}|x^t), \quad (18)$$

where $\sigma_t$ is the standard deviation of original distribution $p_\theta(x^{t-1}|x^t)$ that we already have. Therefore, we only need to solve $\nabla_{x^t} \log p(x_{adv}|x^t)$.

$p(x_{adv}|x^t)$ can be interpreted as the probability that $x^t$ will eventually be denoised to a original image close to $x_{adv}$. Then, a heuristic formula can be used to approximate the probability distribution:

$$p(x_{adv}|x^t) = \frac{1}{Z} \exp(-sD(x^t, x_{adv}^t)), \quad (19)$$

Table 5: Datasets partitioning for uni-modal retrieval models

| Datasets | Total | Training | Query | Database | Class |
|---|---|---|---|---|---|
| FLICKR-25K | 25000 | 5000 | 1700 | 23300 | 38 |
| MS-COCO | 122218 | 10000 | 5000 | 117218 | 80 |
| NUS-WIDE | 195834 | 10500 | 2100 | 193734 | 21 |

where $D$ is some distance metric, $Z$ is a normalization, $s$ is a scale factor that controls the magnitude of the guidance, and $x_{adv}^t$ is obtained by diffusion $x_{adv}$ $t$ steps. When the purified image $x^t$ to be close to the adversarial image $x_{adv}^t$ in the reverse process, $D(x^t, x_{adv}^t)$ will be smaller. When $D(x^t, x_{adv}^t)$ is smaller, probability $p(x_{adv}|x^t)$ is larger, and the purified $x^t$ is closer to the adversarial sample $x_{adv}$.

Here we take the logarithm of both sides and then calculate the gradient of them:

$$\log p(x_{adv}|x^t) = -\log Z - sD(x^t, x_{adv}^t), \tag{20}$$

$$\nabla_{x^t} \log p(x_{adv}|x^t) = -s\nabla_{x^t} D(x^t, x_{adv}^t), \tag{21}$$

By combining formula 18 and formula 21, the expression for the conditional gradient can be obtained:

$$\sigma_t^2 g = -s\sigma_t^2 \nabla_{x^t} D(x^t, x_{adv}^t). \tag{22}$$

## D    DETAILS OF DATASETS SETTINGS

Followed by the studies Wang et al. (2021b) and Wang et al. (2023), we split each dataset into a queryset and database, and part of the database is randomly selected as the training set. The training set is used to optimize the attacked hashing retrieval models and attack methods, while the query set is first used to generate adversarial examples and then evaluate the targeted attack together with the database set. To avoid accidental results caused by using a uniform target label, as in existing works (Wang et al. (2021b), Bai et al. (2020b)), we set different target labels for each benign example, where these target labels are randomly sampled from datasets.

Hash retrieval can be divided into two types: one type is a uni-modal retrieval model, which inputs query images and outputs semantically similar images in the database. Another type is a cross-modal hashing retrieval model, which has four representations of input and output: Image to Image (I2I), Image to Text (I2T), Text to Image (T2I), Text to Text (T2T). We conducted experiments on the attack methods of these two types of retrieval models, and due to the different retrieval tasks, we will discuss the differences in the versions and partitions of the datasets.

1) Uni-modal retrieval task datasets: As shown in Table 5, we adopt the same experimental dataset and partitioning settings as Wang et al. (2021b).

2) Cross-modal retrieval task datasets: As shown in Table 6, we adopt the same experimental dataset and partitioning settings as Wang et al. (2023).

## E    DETAILS OF ATTACKED METHODS AND ATTACK ATTACKS

**1) Uni-modal retrieval task** In the uni-modal hash retrieval task, we use three targeted attack methods to generate adversarial samples and purify them to test the improvement of retrieval performance. Among them, P2P and DHTA are iterative attacks (Bai et al. (2020b)), while ProS-GAN (Wang et al. (2021b)) is generative attack. Iterative methods are inefficient because they are optimization-based methods that rely on a very timeconsuming iterative gradient. Unlike the former, generative attacks are an efficient and effective attack in deep hash based retrieval. By using the original image and target label as inputs, it can generate targeted adversarial examples that can mislead attack hash networks to retrieve images related to the semantics of the target label.

Table 6: Datasets partitioning for uni-modal retrieval models

| **Datasets** | Total | Training | Query | Database | Class |
|---|---|---|---|---|---|
| Wikipedia | 2866 | 2173 | 693 | 2173 | 10 |
| IAPR-TC12 | 20000 | 5000 | 2000 | 18000 | 255 |
| FLICKR-25K | 20015 | 5000 | 2000 | 18015 | 24 |
| MS-COCO | 123287 | 10000 | 5000 | 121287 | 80 |
| NUS-WIDE | 195834 | 10500 | 2100 | 193734 | 21 |

We trained the ProS-GAN attack method on three datasets using the official repository (https://github.com/xunguangwang/ProS-GAN). This method consists of three parts: a Prototype-Net, a Decoder and a Generator. For the network architecture, we built PrototypeNet with four-layer fully-connected networks ($y_t \rightarrow 4096 \rightarrow 512 \rightarrow r_t \rightarrow y_t, h_t$). We set $\alpha_1$, $\alpha_2$ and $\alpha_3$ as 1, $10^{-4}$ and 1. We adopt a fully-connected layer and four deconvolutional layers for the Decoder $D_t$ to upsample the semantic representation $r_t$. We adapt the architecture for Generator $G_{xt}$ from Zhu et al. (2017), and the discriminator contains five stride-2 convolutions and last layer with a $7 \times 7$ convolution. The weighting factor $\alpha$ are set with 50 for NUS-WIDE and MS-COCO, 100 for FLICKR-25K, and $\beta$ is set as 1. For the optimization procedure, we use Adam optimizer with initial learning rate $10^{-4}$. The training epochs are 100 in batch size 24.

After training ProS-GAN, we will use the PrototypeNet and the Generator to attack the attacked hashing network. we select the objective function of DPSH (Li et al. (2015)) and DPH (Cao et al. (2018)) as default method, which are two representative deep hashing methods, to construct the target hashing model. Specifically, VGG-11 (Simonyan & Zisserman (2014)) is adopted as the default backbone network. We replace the last fully connected layer of VGG-11 with the hashing layer, including a new fully-connected layer and the Tanh activation.

### 2) Cross-modal retrieval task

In cross-modal retrieval task, the focus is on image to image and image to text conversion. TA-DCH (Wang et al. (2023)) is the only target attack against cross-modal hash retrieval models. We conducted sufficient experiments on five datasets. Firstly, TA-DCH attacked the images in the query set, and then input these images into the purifier to counteract noise. Finally, we tested the retrieval performance on two query systems.

We trained the TA-DCH attack method on five datasets using the official repository (https://github.com/tswang0116/TA-DCH). Similar to the ProS-GAN method, the structure of the TA-DCH attack method also includes a PrototypeNet, a Decoder, and a Generator. The hyper-parameters $\alpha$, $\beta$, $\mu$, $\nu$ and $\epsilon$ are set to 1, 5, 5, 1 and $*$ respectively, where $* \in [1, 5000]$ is determined according to the given attacked hashing network. For the attacked models that have no specific image feature extractor, we use CNN-F Chatfield et al. (2014) to obtain 4096-dimensional image features. We adopt the Adam optimizer with learning rate $10^{-4}$ to optimize the TA-DCH method. For the training configuration, the two stages of our method have different settings: In the cross-modal prototype learning stage, the training epoch and batch size are set to 20 and 64, respectively. In the adversarial generation stage, the training epoch and the batch size are set to 100 and 24, respectively.

To verify the effects of the proposed model on attacking different deep cross-modal hashing methods, we select two representative ones (Yu et al. (2021);Bai et al. (2020a)) as the attacked objects, including Deep Adversarial Discrete Hashing (DADH) and Deep Graph-neighbor Coherence Preserving Network (DGCPN). Among them, DADH is supervised method, and DGCPN is unsupervised. Since our attack objects are the hashing models that achieve excellent retrieval performance, we train these attacked models separately to fit each dataset. All the attacked models use the unified data pre-processing and follow the setups of their original papers. We find that DADH could not be properly optimized on MS-COCO due to gradient explosion, so we moderately modify its loss function to make it trainable.

It is worth noting that both ProS-GAN and TA-DCH are generative attack methods, and the structure is composed of three sub networks: PrototypeNet, adversarial generator network, and discriminator network. Since our attack objects are the hashing models that achieve excellent retrieval performance, we train these attacked models separately to fit each dataset. All the attacked models use the unified data pre-processing and follow the setups of their original papers.

## F    THE RESULTS OF PURIFY ORIGINAL DATA

To demonstrate the impact of the purification model on original data, we conducted comparative experiments in both uni-modal and cross-modal hash retrieval systems. As shown in the Table 7, Table 9 and Table 9, by comparing the retrieval results of purified original data and the original data in the retrieval model, we can conclude that the purification model has a minor impact on the original data, and the retrieval results are close to those of the original data retrieval.

Table 7: Purify original datasets of uni-modal retrieval task.

| I2I | | FLICKR-25K | | MS-COCO | | NUS-WIDE | |
|---|---|---|---|---|---|---|---|
| | | DPSH | DPH | DPSH | DPH | DPSH | DPH |
| MAP | original | 81.06 | 77.07 | 63.69 | 58.95 | 73.06 | 69.92 |
| | **RetPur** | **80.02** | **76.57** | **62.85** | **58.46** | **72.76** | **69.82** |
| T-MAP | original | 73.67 | 73.21 | 50.53 | 49.22 | 59.17 | 60.27 |
| | **RetPur** | **71.71** | **72.35** | **46.30** | **45.99** | **53.53** | **53.56** |

Table 8: Purify original datasets of cross-modal retrieval task for Image to Image (I2I).

| I2I | | WIKI | | IAPR | | FLICKR-25K | | MS-COCO | | NUS-WIDE | |
|---|---|---|---|---|---|---|---|---|---|---|---|
| | | DGCPN | DADH | DGCPN | DADH | DGCPN | DADH | DGCPN | DADH | DGCPN | DADH |
| MAP | original | 38.09 | 44.93 | 64.74 | 69.35 | 81.99 | 80.58 | 61.92 | 58.49 | 77.91 | 75.60 |
| | **RetPur** | **36.98** | **44.83** | **63.57** | **67.10** | **80.71** | **82.99** | **60.20** | **56.24** | **76.82** | **74.66** |
| T-MAP | original | 13.33 | 12.01 | 31.17 | 33.43 | 56.71 | 57.07 | 34.27 | 36.91 | 35.45 | 35.92 |
| | **RetPur** | **14.24** | **13.30** | **31.18** | **31.09** | **56.10** | **54.94** | **35.86** | **35.68** | **35.91** | **34.93** |

Table 9: Purify original datasets of cross-modal retrieval task for Image to Text (I2T).

| I2T | | WIKI | | IAPR | | FLICKR-25K | | MS-COCO | | NUS-WIDE | |
|---|---|---|---|---|---|---|---|---|---|---|---|
| | | DGCPN | DADH | DGCPN | DADH | DGCPN | DADH | DGCPN | DADH | DGCPN | DADH |
| MAP | original | 37.79 | 43.15 | 65.24 | 72.49 | 80.56 | 78.45 | 65.21 | 62.28 | 76.42 | 77.65 |
| | **RetPur** | **36.73** | **42.93** | **64.17** | **71.03** | **79.33** | **85.30** | **63.25** | **59.94** | **75.32** | **76.61** |
| T-MAP | original | 13.34 | 12.11 | 31.37 | 37.60 | 56.54 | 62.39 | 34.25 | 38.67 | 35.64 | 39.23 |
| | **RetPur** | **14.18** | **13.37** | **31.33** | **35.81** | **56.34** | **60.43** | **36.63** | **37.06** | **35.84** | **38.28** |

