# OpenReview forum: "RetPur: Diffusion Purification Model for Defending Hash Retrieval Target Attacks"
_ICLR.cc/2024/Conference — Submitted to ICLR 2024_

### Official Review · Reviewer_vSmf · 2023-10-26

**Soundness:** 3 good
**Presentation:** 2 fair
**Contribution:** 3 good
**Rating:** 6
**Confidence:** 4

**Summary:**

The authors introduce a purification model named "RetPur" designed to purify adversarial test datasets. This helps mitigate targeted attacks in both uni-modal and cross-modal retrieval systems.
The "RetPur" model uses a pre-trained diffusion model, which provides plug-and-play convenience. It also utilizes adversarial samples as conditioning factors to guide image generation, enhancing task accuracy.
The study pioneers the incorporation of adversarial purification tasks into uni-modal (Image-to-Image) and cross-modal (Image-to-Image, Image-to-Text) hash retrieval systems, especially tailored for image retrieval scenarios.
The authors explore the application of adversarial purification tasks against a variety of attacks, including both generative and iterative approaches.

**Strengths:**

The paper showcases originality by addressing a novel problem in the domain of image retrieval, maintains a high standard of quality in its methodology and comparative analysis, is presented with clarity, and holds significant implications for the broader field of deep learning and image retrieval.​

**Weaknesses:**

Potential Overhead and Scalability: The paper introduces a plug-and-play feature without addressing the potential computational overhead or scalability when integrating "RetPur" into existing retrieval systems.
Actionable Insight: An evaluation of the computational cost, in terms of time and resources, when deploying "RetPur" would be valuable. The authors should discuss the feasibility of using their model in real-world, large-scale applications and any potential bottlenecks.

**Questions:**

Are there specific types of adversarial attacks or scenarios where "RetPur" might not be as effective?

---

### Official Review · Reviewer_6D5x · 2023-10-30

**Soundness:** 2 fair
**Presentation:** 3 good
**Contribution:** 1 poor
**Rating:** 3
**Confidence:** 4

**Summary:**

This paper purifies datasets to defend the attacks in image retrieval models. Following the previous methods, based on a pre-trained diffusion model, the authors utilize adversarial examples as the conditioning factor to guide image generation. They attack both uni-modal and cross-modal retrieval tasks. The results demonstrate that the proposed method could improve the retrieval performance.

**Strengths:**

This paper is well-written and easy to understand.
The proposed method is effective in improving the defense performance in their experiments.

**Weaknesses:**

Considering the existing works [1][2], I think that the contribution of this paper is limited. The authors also utilize the adversarial examples as guidance [1] and leverage the forward and reverse diffusion process to remove the perturbations [2]. This paper transfers this protection paradigm from image classification to image retrieval tasks. Therefore, I think the novelty of this paper is below the acceptance threshold.

[1] Jinyi Wang, Zhaoyang Lyu, Dahua Lin, Bo Dai, and Hongfei Fu. Guided diffusion model for adversarial purification. arXiv preprint arXiv:2205.14969, 2022.
[2] Weili Nie, Brandon Guo, Yujia Huang, Chaowei Xiao, Arash Vahdat, and Anima Anandkumar. Diffusion models for adversarial purification. arXiv preprint arXiv:2205.07460, 2022.

**Questions:**

The proposed method is not specific to image retrieval tasks or cross-modal relationships. Why not conduct experiments on other tasks?

---

### Official Review · Reviewer_o7o6 · 2023-10-31

**Soundness:** 3 good
**Presentation:** 3 good
**Contribution:** 2 fair
**Rating:** 3
**Confidence:** 4

**Summary:**

The paper introduces RetPur, a novel purification model designed to enhance the robustness of hash retrieval systems against targeted attacks by purifying adversarial test datasets. RetPur employs a pre-trained diffusion model and uses adversarial samples as conditioning factors to guide image generation, thereby improving task accuracy. The contributions of this model lie in its effectiveness in mitigating the impact of adversarial attacks and its applicability to both uni-modal (image-to-image) and cross-modal (image-to-image, image-to-text) hash retrieval systems.

**Strengths:**

1.The authors introduce adversarial purification into hash retrieval systems, mitigating the issue of targeted attacks within both uni-modal and cross-modal retrieval systems by purifying adversarial test datasets.

2.The authors have validated the effectiveness of the proposed RetPur purification model through extensive experiments and have explored the application of adversarial purification tasks in a wider range of attack scenarios.

**Weaknesses:**

1.The authors did not provide a specific explanation of the differences between the proposed RetPur and the 'Guided Diffusion Model for Adversarial Purification' article's purification methods.

2.The novelty of the method may be somewhat lacking; in my opinion, the purification method is akin to preprocessing the data, and the preprocessed clean data can evidently be used for various downstream tasks.

3.In the paper, there are several typo errors , for instance, in Figure 1, 'RetPur' is incorrectly written as 'RerPur,' and in the 'CONCLUSION' section, the model name 'RetPur' is mistakenly written as 'RerPur' and 'PerPur.'

**Questions:**

1.Could the authors provide a detailed explanation of the differences between their RetPur method and the purification method in the 'Guided Diffusion Model for Adversarial Purification' article?

2.Could the authors provide more ablation experiments using different datasets, attack methods, and retrieval models?

---

### Meta-Review · Area_Chair_9xYU · 2023-12-08

**Metareview:**

The submission proposes a purification model to improve the robustness of hash retrieval systems towards targeted attacks, termed RetPur. The reviewers provided mixed scores for the submission, seeing on the one hand the benefit of the proposed model, yet, raising concerns on the delineation from prior work, specifically, Wang et al: Guided Diffusion Model for Adversarial Purification, that is quite related, leading to concerns w.r.t. novelty. Further, there are concerns regarding the methods scalability.
There was no rebuttal provided.

**Justification For Why Not Higher Score:**

The reviews raise novelty issues and opt for rejection. No rebuttal was provided.

**Justification For Why Not Lower Score:**

N/A

---

### Decision · Program_Chairs · 2024-01-16

Reject